# Effects of Iron Supplementation on Testicular Function and Spermatogenesis of Iron-Deficient Rats

**DOI:** 10.3390/nu14102063

**Published:** 2022-05-14

**Authors:** Chih-Wei Tsao, Yuan-Ru Liao, Ting-Chia Chang, Yih-Fong Liew, Chin-Yu Liu

**Affiliations:** 1Division of Urology, Department of Surgery, Tri-Service General Hospital, National Defense Medical Center, Taipei 11490, Taiwan; weisurger@gmail.com; 2Division of Experimental Surgery Center, Department of Surgery, Tri-Service General Hospital, National Defense Medical Center, Taipei 11490, Taiwan; 3Department of Nutritional Science, Fu Jen Catholic University, New Taipei City 24205, Taiwan; weisurger228@yahoo.com.tw (Y.-R.L.); ctc5628@gmail.com (T.-C.C.);

**Keywords:** iron deficiency, iron supplementation, testis, spermatogenesis, oxidative stress

## Abstract

Iron deficiency is the most common micronutrient deficiency in the world. Previous studies have shown that iron deficiency increases oxidative stress and decreases antioxidant enzymes, and studies of male infertility indicated that oxidative stress may affect male reproductive functions. The aim of this study was to investigate the effects of iron supplementation on spermatogenesis and testicular functions in iron-deficient rats. Three-week-old male Sprague Dawley (SD) rats were randomly divided into two groups: an iron-adequate control (AI group, 35 ppm FeSO_4_) and an iron-deficient group (ID group, <5 ppm FeSO_4_). After three weeks, the iron-deficient group was divided into an original iron-deficient group and five iron-supplemented groups, the latter fed diets containing different doses of FeSO_4_ (6, 12, 18, 24, and 35 ppm). After five weeks, blood and testis tissue were analyzed. We presented as median (interquartile range, IQR) for continuous measurements and compared their differences using the Kruskal–Wallis test followed by the Mann–Whitney U test among groups. The results showed that as compared with the AI group, the ID group had significantly lower serum testosterone and poorer spermatogenesis (The medians (QR) were 187.4 (185.6–190.8) of AI group vs. 87.5 (85.7–90.4) of ID group in serum testosterone, *p* < 0.05; 9.3 (8.8–10.6) of AI group vs. 4.9 (3.4–5.4) of ID group in mean testicular biopsy score (MTBS], *p* < 0.05); iron supplementation reversed the impairment of testis tissue. In the testosterone biosynthesis pathway, iron supplementation improved the lowered protein expressions of hydroxysteroid dehydrogenases caused by iron deficiency. Additionally, decreased activities of glutathione peroxidase and catalase, and increased cleaved-caspase 8 and caspase 3 expression, were found in the iron-deficient rats. The iron-supplemented rats that received > 12 ppm FeSO_4_ exhibited improvements in antioxidant levels. In conclusion, iron supplementation can abrogate testis dysfunction due to iron deficiency through regulation of the testicular antioxidant capacity.

## 1. Introduction

Iron deficiency is the most common micronutrient deficiency according to published statistics, and has been reported to be frequently found in anemic patients, with a prevalence of up to 50% [1,2]. In addition, iron-deficiency anemia was reported to be one of the five leading causes of years lived with disability (YLDs) in 2016 [3]. Briefly, iron deficiency can be categorized into an increased iron requirement, inadequate iron intake, and decreased iron absorption. Populations such as infants, children, and adolescents during the growth spurt, women during menstruation, and pregnant and lactating women have greater iron requirements. An inadequate iron intake may result from appetite loss, a vegan dietary pattern, or low dietary quality with poor iron sources. Unlike heme iron, an animal source of iron, non-heme iron obtained from plants has a lower bioavailability. In addition, surgery, Crohn’s disease, side effects of drugs, and Helicobacter pylori infection cause malabsorption of iron [4], and increased iron loss associated with bleeding due to surgery, trauma, or blood donation tends to lead to the development of iron deficiency [5].

Chronic or severe iron deficiency affects the oxygen-carrying ability of red blood cells and thus causes hypoxia, which has been found to increase the risks of acute coronary syndrome and myocardial infarction [6]. In addition, evidence shows that iron deficiency in childhood leads to poorer brain neurophysiologic development [7]. Cognitive impairments and memory deficits due to alteration of the hippocampus are also adverse effects of iron deficiency [8].

In the male reproductive system, the testes maintain testosterone and sperm production via the regulation of Sertoli cells, Leydig cells, and upstream hormones involved in the hypothalamic-pituitary-gonadal axis [9]. Ferritin, an easily-available iron source, existing in Sertoli and Leydig cells helps boost sperm production and protect the testes [10]. Oxidative stress has been demonstrated to be a major cause of male reproductive dysfunction [11,12]. Imbalance in the redox status requires enzymatic and non-enzymatic antioxidants to diminish. Iron deficiency affects iron-porphyrin enzymes such as peroxidases and catalase. The results of a human study including 33 female iron-deficient anemia patients performed in order to analyze the redox status showed that the activity of catalase and the total antioxidant capacity in the anemia group were significantly lower than in the controls. Moreover, the oxidant activity was elevated in the anemia group [13]. Blocking of defense mechanisms causes the generation of reactive oxygen species (ROS) and increases oxidative stress, leading to mitochondrial, DNA and other cellular damage, thus inducing apoptosis [11,14]. Nagababu et al. reported that iron-deficient diet-fed mice had a greater erythrocytic oxidative stress, as iron-deficiency anemia reduced the lifespan of blood cells, causing hemoglobin autoxidation and ROS production [15].

Human fertility is affected not only biological consequences but also so socioeconomic factors [16]. Declining fertility is a key driver behind the rapid aging of populations worldwide. Taiwan is also going on the demographic transition from high birth and death rates to low birth and death rates. Gender-dependent differences are one of the issues but women always take on a disproportionate share of fertility tasks. Unlike female infertility, male infertility is not well reported in general. Thus, we want to explore the association between male reproductive health-related diseases and the potential role of nutrition factors.

The present study aimed to explore the effects of iron deficiency on rat testis functions, ascertain whether any improvement occurred in iron-deficient rats after iron supplementation, and examine whether a dose-response relationship exists by comparing rats receiving different doses of iron supplementation.

## 2. Materials and Methods

### 2.1. Animals and Diets

Fifty-two three-week-old male Sprague Dawley (SD) rats were purchased from BioLASCO Taiwan Co., Ltd. (Taipei, Taiwan), housed in together but kept in the cage individually under controlled conditions (room temperature 22–24 °C, humidity 50–60%, 12-h light/12-h dark cycle) with free access to water and food. After adaptation, the rats were divided into a control group (AI group, *n* = 6) and an iron-deficient group (ID group, *n* = 46) and maintained for 3 weeks to create an iron-deficient rat model. The iron-deficient rats were then randomly allocated into six groups as follows: iron-deficient group (ID group, *n* = 8) and five iron-supplemented groups (S6, S12, S18 and S24 groups, *n* = 8; S35 group, *n* = 6) for 2 weeks. All diets were modified from the AIN-93G rodent standard diet (#115072, Dyets, Inc., Bethlehem, PA, USA). For the AI group, 35 ppm sulfate heptahydrate (FeSO_4_·7H_2_O, F7002, Sigma-Aldrich, Saint Louis, MO, USA) was added to the AIN-93G diet, while less than 5 ppm FeSO_4_ was added to the AIN-93G diet for the ID group. For the iron-deficient rats supplemented with iron in the 5 experimental weeks, 6, 12, 18, 24, and 35 ppm FeSO_4_ were added to the AIN-93G diet for the S6, S12, S18, S24, and S35 groups, respectively. The rats were finally fasted for 12 h and euthanatized by inhalation exposure to carbon dioxide for at least 5 min. Blood and testes samples were collected for further analyses.

### 2.2. Hematology Analyses

The hemoglobin (Hgb) content in whole blood was measured using a commercial Hgb kit (HG1539, Randox, Crumlin, UK) according to the manufacturer’s instructions. For cholesterol and testosterone measurement, samples of serum were collected and assayed by the Taiwan Society of Laboratory Medicine (Taipei, Taiwan). The cholesterol level was determined by the enzymatic (CHOD-POD) method, and the testosterone level was measured using a Centaur XP (0-SM-CENTAURXP, Siemens Healthcare, Erlangen, Germany).

### 2.3. Hematoxylin and Eosin Staining

Testis tissue was soaked in 10% formaldehyde solution (11-0735, Sigma-Aldrich, Saint Louis, MO, USA) for 24 h, then dehydrated, embedded, sliced to 3 μm, and mounted on glass slides at the Department of Pathology of Cardinal Tien Hospital (New Taipei City, Taiwan). The slides were processed using a Leica Autostainer XL (ST5010, North Chicago, IL, USA) and mounted with coverslips. The stained slides were observed under a light microscope (Eclipse E400, Nikon, Tokyo, Japan) at ×400 magnification. The mean seminiferous tubule diameter (MSTD) was measured and the mean testicular biopsy score (MTBS) [17] was ascertained.

### 2.4. Immunofluorescence Staining

For immunofluorescence staining, slides were placed in an oven at 65 °C for 4 h and soaked in Histo-Clear II (National Diagnostics, Atlanta, GA, USA) overnight. Then, slides were placed in Histo-Clear II for 10 min twice, followed by 100%, 95%, 80%, and 70% ethanol for 30 s once, and rinsed with dH_2_O for 10 min once and washed with 1× TBS (Bioman, New Taipei City, Taiwan) for 5 min twice. Slides were subsequently immersed in Lectin solution (L21409, Thermo Fisher Scientific, Waltham, MA, USA) for 20 min, followed by 1× TBS wash for 5 min, DAPI solution (4′,6-diamidino-2-phenylindole; D3541, Thermo Fisher Scientific) for 3 min, and 1× TBS wash for 5 min, then mounted with coverslips. Slides were observed under a light microscope (Eclipse E400, Nikon) to determine the stages of testicular seminiferous tubules.

### 2.5. TUNEL Staining

Terminal deoxynucleotidyl transferase dUTP nick end labeling (TUNEL) staining was performed using the DeadEndTM Colorimetric TUNEL System (G7130, Promega, Madison, WI, USA). In brief, 3-μm paraffin-embedded slides were soaked in 0.85% saline for 5 min followed by 1× PBS wash for 5 min. Then, 100 μL Proteinase K solution (20 μg/mL) was added to the slides for 15 min, which were then washed with 1× PBS three times (5 min each), followed by the addition of 100 μL equilibration solution. After 10 min, 100 μL rTdT solution was added and coverslips placed on the slides for 60 min in a 37 °C water bath. The coverslips were then removed, and 2× SSC solution was added for 15 min, followed by 1× PBS wash three times (5 min each), HRP solution for 30 min, and 1× PBS wash three times (5 min each). Subsequently, 100 μL DAB solution was added for 1 min, and slides were then rinsed with dH_2_O for 3 min. Slides were observed under a light microscope (Eclipse E400, Nikon) to determine changes to the nuclei of apoptotic cells in the testis.

### 2.6. Testicular Redox Status

The testicular redox status, including the activities of superoxide dismutase (SOD), glutathione peroxidase (GPx), and catalase (CAT), was assessed. Testis tissue (about 0.05 g) was added to ice-cold 200 μL RIPA buffer, 2 μL protease inhibitor, and 2 μL phosphatase inhibitor, homogenized, and centrifuged at 14,000× *g* for 15 min at 4 °C, following which the supernatant was transferred to a new 1.5-mL Eppendorf flask and stored at −80 °C. All parameters were measured using the testicular lysate following protocols provided with the assay kits (706002, Cayman, Ann Arbor, MI, USA; 703102, Cayman; 707002, Cayman).

### 2.7. Western Blotting

Protein extraction was performed on testis tissue using the methods described in the former paragraph. Samples containing 20 μg of protein were loaded on a 10% sodium dodecylsulfate polyacrylamide gel electrophoresis (SDS-PAGE) gel and transferred to polyvinylidene difluoride (PVDF) membranes (Bioman) via wet transfer. After soaking in blocking buffer (5% non-fat milk dissolved in 1× TBST) for 1 h, membranes were incubated with primary antibodies overnight at 4 °C. The following primary antibodies were used for testosterone biosynthesis: CYP11A1 (1:1000, sc-292456, Santa Cruz, CA, USA), CYP17A1 (1:1000, sc-66850, Santa Cruz), 3β-HSD (1:500, sc-28206, Santa Cruz), 17β-HSD (1:250, sc-135044, Santa Cruz), StAR (1:1000, sc-25806, Santa Cruz); for three forms of SOD: SOD1 (1:1000, ab16831, Abcam, Boston, MA, USA), SOD2 (1:1000, ab16956, Abcam), SOD3 (1:1000, ab83108, Abcam); for the apoptosis pathway: cleaved-caspase 8 (1:1000,GTX59607, GeneTex, Irvine, CA, USA), Bcl-XL (1:1000, ab32370, Abcam), Bax (1:1000, #2772, Cell Signaling, Danvers, MA, USA), caspase 9 (1:1000, #9508, Cell Signaling), caspase 3 (1:1000, #9662, Cell Signaling), PARP (1:1000, #3542, Cell Signaling); and for loading control: β-actin (1:10,000, A5316, Sigma-Aldrich). After three washes with 1× TBST, membranes were then incubated with either goat anti-mouse IgG secondary antibody (1:5000, sc-2005, Santa Cruz) or goat anti-rabbit IgG secondary antibody (1:5000, sc-2054, Santa Cruz) for 1 h. Following three washes with TBST, the bands on the PVDF membranes interacted with ECL solution (1705061, Bio-Rad, Santa Rosa, CA, USA) and were detected by densitometry.

### 2.8. Statistical Analyses

The experimental results were analyzed using SAS Software version 9.4 (SAS Institute Inc., Cary, NC, USA) and presented as median (interquartile range, IQR) because of non-normal distribution. Data were compared using the Kruskal–Wallis test followed by the Mann–Whitney U test; statistical significance was considered when *p* < 0.05. In addition, Spearman’s correlation test was conducted to estimate linear regression between two variables.

## 3. Results

### 3.1. Body Weight and Food Intake

Before receiving an iron-deficient diet, the rats in the AI group and the ID group had a similar mean body weight. However, after 3 weeks of an iron-deficient diet, the ID group had less weight gain and a lower food intake as compared with the AI group. A two-week period of iron supplementation significantly increased the body weight and food intake in all iron-supplemented groups as compared with the ID group; the content of iron in the diet was positively associated with food intake and thus increased body weight. Data are presented in Figure 1.

### 3.2. Biochemical Values

The iron-deficient diet caused a reduction in the blood Hgb level, but this level was elevated in a dose-dependent manner after iron supplementation; however, all iron-supplemented groups had a significantly lower Hgb level than the control group. In addition, the serum testosterone concentration of the rats fed an iron-deficient diet showed a significant reduction in comparison with the rats fed an iron-adequate diet. However, the iron-deficient rats treated with different doses of iron displayed a gradual increase in testosterone concentration, but only the rats in group S35 reached a normal testosterone concentration. Serum cholesterol was measured after the experiment, as it serves in the role of testosterone precursor; however, the levels of cholesterol in all groups were similar (Figure 2a). Statistical comparisons of three values revealed a positive correlation (R = 0.58) between the levels of serum testosterone and serum Hgb (Figure 2b), from which the hypothesis that iron status may not influence the cholesterol level, but does influence the blood Hgb and testosterone levels, may be derived.

### 3.3. Protein Expressions of Testosterone Biosynthesis Enzymes

As the precursor content was similar in all groups, enzymes involved in testosterone biosynthesis were measured using western blot (Figure 3). In comparison with the AI group, the ID group had a lower StAR protein expression and a higher 3β-HSD protein expression. After iron supplementation, the StAR protein expression was increased, and a decreasing trend in 3β-HSD was observed only in group S35. In addition, the iron-supplemented groups displayed a gradual increase in 17β-HSD; moreover, groups S12, S18, and S24 had significantly higher protein expressions than the AI and ID groups, and group S35 had higher protein expressions than the ID group (Figure 3).

### 3.4. Testicular Histology

As shown in the figures, unlike the normal structure observed in the AI group, testicular sections from rats exposed to a low-iron diet exhibited a thinner seminiferous epithelium, cavities, and scattered and decreased germ cells in seminiferous tubules; iron supplementation improved this damage (Figure 4a). Quantitative analyses of morphological alterations revealed a slightly increased mean seminiferous tubule diameter (MSTD) and a significantly decreased mean testicular biopsy score (MTBS) in the iron-deficient rats, which were reversed in the iron-supplemented rats. The former parameter, MSTD, was reduced in groups S6, S12, and S24, but no significant difference was seen in groups S18 and S35, while the latter parameter, MTBS, was elevated in all supplemented groups, and no difference was observed as compared with the controls in the S24 and S35 groups (Figure 4b).

### 3.5. Testicular Spermatogenesis

Testicular sections were stained with nuclear marker DAPI and acrosome marker lectin to detect stages of spermatogenesis and types of spermatogenic cells in the different stages. In reference to Nakata et al. [18], five stages (stages II–III to VI) of spermatogenesis were determined in the present study. The lectin-stained shape of the acrosome of spermatids was dot-like in stages II-III (Figure 5a) and larger in stage IV (Figure 6a); the acrosome became spindle-shaped in stage V (Figure 7a), and turned triangle-shaped in stage VI (Figure 8a). Fluorescence images of rat seminiferous tubules in stages II-III and stage IV showed greater numbers of germ cells in the ID group as compared with the AI group, while in stages V and VI, more germ cells were observed in the AI group; however, no statistically significant difference between the ID and AI groups was observed at any stage. In addition, the iron-deficient rats exhibited an increasing trend in the number of germ cells after iron supplementation in stages II-III, IV, and VI. In detail, germ cells in stage IV in group S6 were significantly lower in number than in group S18 and the ID group; in stage V, group S18 had an apparently higher number of germ cells than groups S12, S24, S35, and the ID group; and in stage VI, germ cells in group S18 numbered higher than in group S6. Data for stages II-III are shown in Figure 5b, stage IV in Figure 6b, stage V in Figure 7b, and stage VI in Figure 8b.

### 3.6. Testicular Redox Status and Protein Expression of SOD

Despite no difference in SOD activity being observed between the AI and ID groups, decreasing activities of GPx and catalase were observed in the iron-deficient rats. After iron supplementation, all parameters exhibited a gradually increasing trend. In detail, the SOD activity in group S18 was higher than in group S12 and the ID group; the GPx activity in group S24 was higher than in groups S18 and S35, and the ID group; and the catalase activities in the groups supplemented with 12 ppm to 35 ppm were higher than in group S6 and the ID group (Figure 9a).

As SOD is a metal-dependent enzyme, the protein expressions of different forms of SOD in the testis were measured using western blot. Iron deficiency resulted in a lower protein expression of SOD2 (Mn-SOD) and higher SOD3 (extracellular Cu/Zn-SOD), whereas a decreasing trend in SOD1 (intracellular Cu/Zn-SOD) was observed. The protein expressions of SOD1 and SOD2 showed a similar trend, increasing in the 6 and 12 ppm iron supplementation groups and decreasing in the 18, 24, and 35 ppm supplementation groups. In contrast, the iron-deficient rats supplemented with doses of iron exhibited marked increases in the protein expression of SOD3 as compared with the rats under a normal iron status. Additionally, the expressions of SOD3 in groups S6, S12, and S24 were significantly higher than in the ID group (Figure 9b).

### 3.7. Testicular Apoptosis and Protein Expressions of Apoptosis Markers

Increased levels of apoptotic cells in testis tissue were observed in the iron-deficient rats using TUNEL analysis, which were reduced after different doses of iron supplementation; however, the levels of apoptotic germ cells in the iron-supplemented groups were significantly higher than in the AI group (Figure 10a,b). The expressions of intrinsic (Figure 11a) and extrinsic (Figure 11b) apoptosis pathway-related markers were determined by western blot. The rats receiving a low-iron diet had a significantly lower expression of caspase 3, and higher expressions of cleaved-caspase 8 and cleaved-caspase 3; however, the iron-supplemented rats did not exhibit any improvements in protein expressions. In detail, the protein expressions of anti-apoptotic marker BCL-xL in groups S6, S12, and S35 were higher than in the AI group, and that in group S12 was higher than in the ID group; in addition, the levels of pro-apoptotic marker Bax in groups S12 and S35 were higher than in the AI and ID groups. Besides, the protein expressions of cleaved-caspase 9, cleaved-caspase 8, and caspase 3 in the iron-supplemented rats were generally higher than those in the iron-deficient and normal rats. Only the change in cleaved-caspase 3 was reversed after 24 ppm iron supplementation.

## 4. Discussion

Few studies have investigated reproductive functions under iron deficiency, and the effects of iron deficiency followed by supplementation with different doses of iron are also unknown. According to the Nutrition and Health Survey in Taiwan from 1993–1996 [19], 2.1% of men was iron deficient, and a higher prevalence of iron deficiency was found in teenage boys and aged men. The reason that the iron-deficient group and five iron-supplemented groups with diets containing different doses of ferrous sulfate were designed, is to closely mimic the different statuses of iron deficiency. In this rat model of diet-induced iron deficiency, reproductive functions including testosterone level, semen quality, testicular histology, and spermatogenesis, in addition to possible modulation pathways, were explored. 

Our study showed that a 3-week low-iron diet (<5 ppm) in rats was capable of reducing the mean Hgb level to 80% lower than that in the control rats, indicating the successful development of an iron-deficient rat model [20]. The iron-deficient rats also had a lesser body weight gain and a lower food intake, in agreement with previous studies [21,22,23]. A 3-week iron-deficient diet-induced rat model developed by Ghada et al. exhibited a lower weight gain, food intake, and Hgb level [24]. Decreased food consumption may be explained by the presence of iron-deficient anemia, which causes fatigue, weakness and lower physical activity owing to a decline in the oxygen-carrying ability of red blood cells [25], thus inducing a lower weight gain. Tanaka et al. reported similar findings, and stated that hypermetabolism in iron-deficient rats may influence the growth rate and lead to a lower body weight [26]. Five-week supplementation with different levels of FeSO_4_, a commonly used treatment for iron deficiency, gradually increased the serum Hgb level, body weight, and food intake, in line with He et al. [23] and Ghada et al. [24]. However, analysis of parameters related to iron deficiency, such as the hematocrit level and erythrocyte count, was lacking in this study; similarly, indicators of iron status, including serum iron level, transferrin saturation, and ferritin, were not measured.

After iron depletion, the serum testosterone level and protein expressions of testosterone production-related enzymes were markedly affected in parallel, which were partly reversed following iron supplementation. A pilot study conducted in 2014 showed an obviously elevated serum testosterone level in anemic men due to iron deficiency after iron administration; moreover, the serum Hgb level was positively correlated with the serum testosterone level [27]. Iron deficiency may lead to hypoxia owing to the low Hgb concentration [28,29], and a deleterious effect of low blood flow on the testes has been reported, in that poor circulation causes hypoxia, increases oxidative stress, and disturbs reproductive functions [30,31].

The serum cholesterol levels of our study were similar among all groups, indicating that iron status did not affect the initial material of testosterone production; however, in terms of the steroidogenic pathway, one of the defective enzymes, StAR, represents the rate-limiting step in testosterone production that regulates cholesterol transport into mitochondria of Leydig cells [32]. Musicki et al. reported that hypogonadism was observed in a sickle-cell anemia mouse model, which could be explained by a decreased protein expression of StAR, which impaired testosterone production via limiting the initial material of testosterone, and may be attributed to over-activation of NADPH oxidase-induced oxidative stress [33]. Furthermore, the iron-deficient and most iron-supplemented rats in the present study showed increasing trends in the protein expressions of hydroxysteroid dehydrogenases, because both 3β-HSD and 17β-HSD require NADPH and NADH in testosterone production [34], whereas NADPH and NADH are related to redox hemostasis [35]. Therefore, the iron status may affect redox homeostasis, regulate the levels of NADPH and NADH, and mediate enzymes involved in testosterone production, thus altering the blood testosterone concentration.

Testosterone plays a pivotal role in spermatogenesis [36], and as mentioned in the former section, iron status affects the blood testosterone concentration. Histochemical stains of the testes were therefore evaluated in order to determine the effects of iron status on testicular histology and spermatogenesis. Iron was found to be an essential factor in spermatogenesis [37], in line with the observations of the testis tissue, which included an increased seminiferous tubule diameter, a thinner seminiferous epithelium, reduced spermatozoa, and disorganized spermatogenesis under the condition of iron deficiency; iron supplementation reversed the damaged testicular histology. Iron supplementation also reversed the impaired epithelial maturation in iron-deficient rats based on Johnsen’s MTBS, which was supported by Sylvester et al. that ferric ions were involved in the formation of spermatogonia and round spermatids [38]. In addition, the results of immunofluorescence staining showed elevated germ cell levels in stages II-IV of spermatogenesis and decreased levels until the administration of 18 ppm iron supplementation. A possible explanation for these results may be owing to divalent metal transporter 1 (DMT1), a factor regulating iron status, being found to increase under iron deficiency [39]. Griffin et al. also reported a higher protein expression of DMT1, a transporter of iron, in seminiferous tubules during spermatogenesis [40]. It appears that iron deficiency leads to increased spermatogonia to stimulate iron absorption, and further studies including an analysis of the expression of DMT1 need to be undertaken.

Studies have revealed that both iron deficiency and iron overload affect redox hemostasis [13,41]. Kurtoglu et al. found that 6-week iron therapy reversed the changes in SOD, CAT, and GPx activities in red blood cells and decreased the plasma MDA content in anemic patients [42]. Akarsu et al. reported that the plasma total antioxidant capacity of patients with iron-deficiency anemia increased after receiving iron therapy [43]. The activities of antioxidative enzymes in this study displayed similar trends or findings to those observed in former studies, suggesting that iron status may mediate testicular antioxidants and affect redox hemostasis. It has been reported that oxidative stress causes greater harmful effects in spermatozoa than in other cells [12,44], whereas both human and animal studies have demonstrated that SOD is present at relatively higher levels than other antioxidative enzymes, protecting spermatozoa via reduction of ROS [11,45,46]. The components of SOD contain metal ions [47], and our results showed that the iron-deficient rats had lower Mn-SOD and higher extracellular Cu/Zn-SOD protein expressions. However, iron supplementation did not result in obvious improvement in types of SOD, but did improve the antioxidant levels, raising questions as to the mechanisms of how iron status affects testicular antioxidants.

Oxidative stress and apoptosis are underlying mechanisms of male infertility, as demonstrated in previous studies [12,48]. In addition, a positive correlation between ROS and apoptosis was reported [14]. Recently, iron deficiency or iron overload was found to enhance sperm oxidative stress, leading to an increase in sperm DNA fragmentation, induction of apoptosis, and finally impairment of semen quality [29,49]. In the present study, iron deficiency-induced apoptosis via upregulating the caspase 8 expression in the extrinsic apoptotic pathway and directly activating caspase 3. A possible explanation may be that testicular mitochondrial ferritin protects mitochondria against oxidative damage by increasing oxygen consumption to reduce ROS production [50]. Hence, it is likely that iron-induced apoptosis is triggered by the extrinsic pathway and not the intrinsic pathway owing to the protective response of mitochondria.

The results of this study demonstrated the potential effects of iron deficiency on testis function and prospective improvements resulting from iron supplementation after iron deficiency. However, a limitation of this study was that parameters such as semen quality and fertility were lacking, and further experiments to assess iron deficiency and reproductive functions are needed. Another limitation was the relatively smaller sample size of the individual group; future studies should take more sample sizes of rats to enroll for the increased statistical power.

## 5. Conclusions

In conclusion, iron deficiency modulated testicular anti-oxidative enzymes and protein expressions of apoptotic markers to affect testosterone biosynthesis-related enzymes and the serum testosterone level, and thus altered spermatogenesis and the morphology of the testis. Under iron supplementation, testicular functions were improved via increasing activities of testicular antioxidants.

## Figures and Tables

**Figure 1 nutrients-14-02063-f001:**
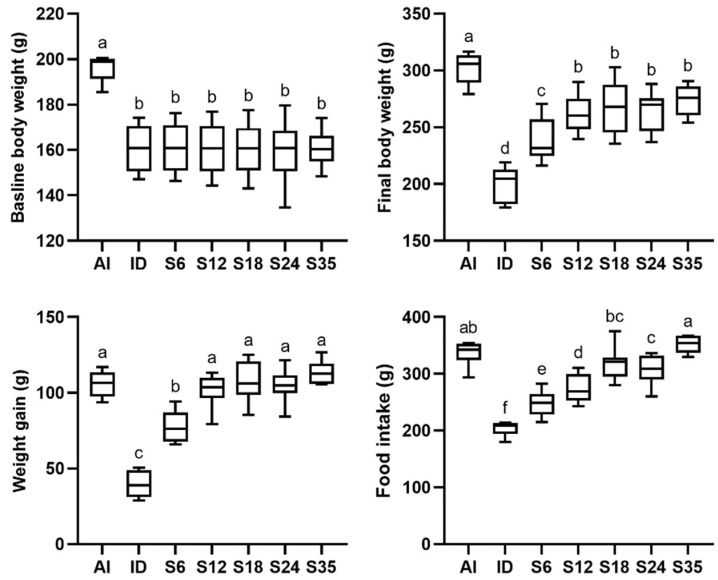
Effects of different doses of iron supplementation on body weight and food intake in iron-deficient rats. Values are presented as median (interquartile range, IQR). *n* = 6–8 per group. Values not sharing the same letters are significantly different at *p* < 0.05. Baseline: the final day of 3-week iron depletion before iron supplementation; Final: after 2-week iron supplementation.

**Figure 2 nutrients-14-02063-f002:**
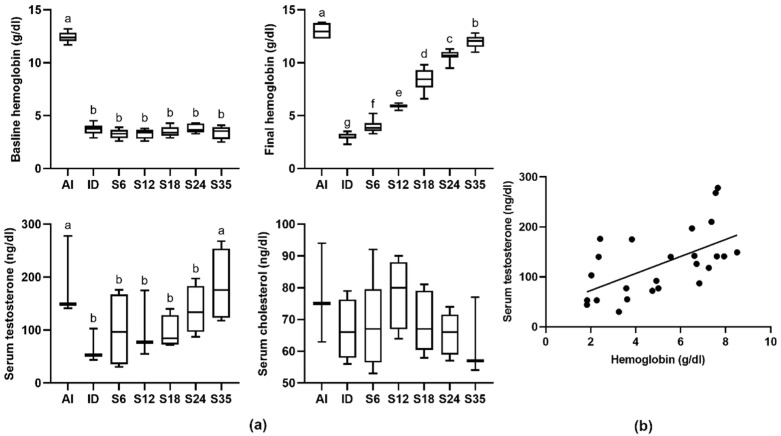
Effects of different doses of iron supplementation on (**a**) blood hemoglobin, testosterone, and cholesterol levels, and (**b**) the relationship between blood hemoglobin and testosterone levels in iron-deficient rats. Values are presented as median (interquartile range, IQR). *n* = 4–8 per group. Values not sharing the same letters are significantly different at *p* < 0.05. Baseline: after 3-week iron depletion and before iron supplementation; Final: after 2-week iron supplementation.

**Figure 3 nutrients-14-02063-f003:**
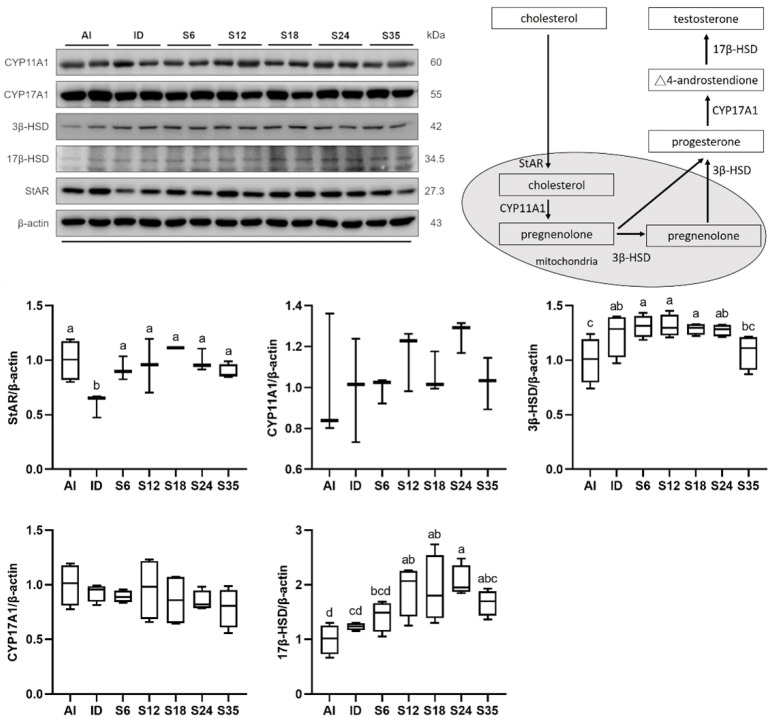
Effects of different doses of iron supplementation on protein expressions of enzymes related to the testicular testosterone biosynthesis pathway in iron-deficient rats. Values are presented as median (interquartile range, IQR). *n* = 3–4 per group. Values not sharing the same letters are significantly different at *p* < 0.05.

**Figure 4 nutrients-14-02063-f004:**
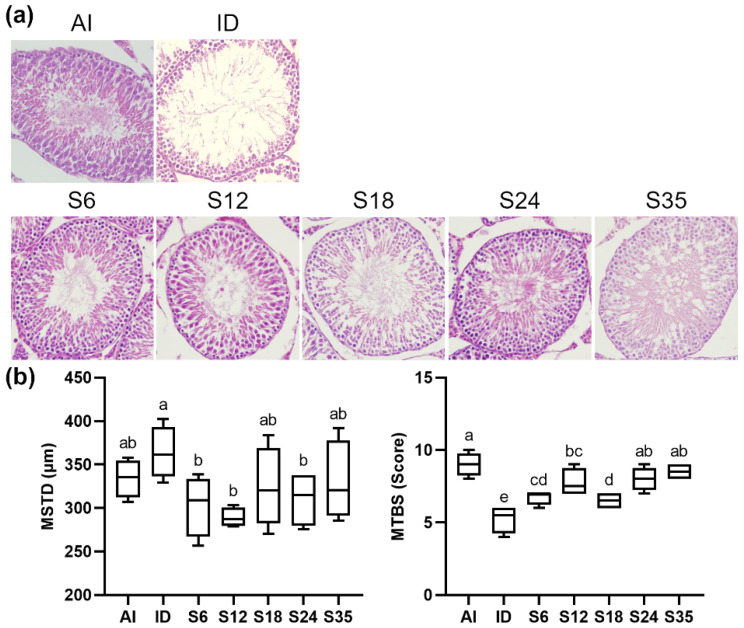
Effects of different doses of iron supplementation on (**a**) testicular histology by H&E staining and (**b**) mean seminiferous tubule diameter (MSTD) and mean testicular biopsy score (MTBS) in iron-deficient rats. Values are presented as median (interquartile range, IQR). *n* = 4 per group. Values not sharing the same letters are significantly different at *p* < 0.05.

**Figure 5 nutrients-14-02063-f005:**
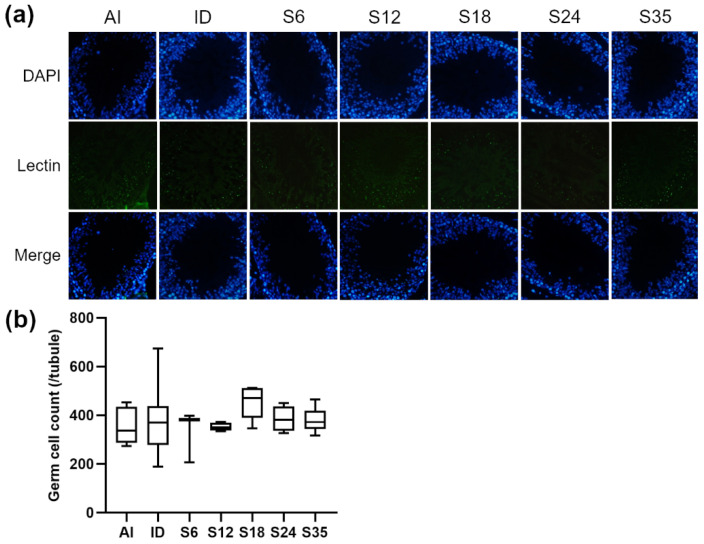
Effects of different doses of iron supplementation on (**a**) testicular seminiferous tubules by DAPI (nucleus) and Lectin (acrosome) staining, and (**b**) germ cell numbers in stages II-III of spermatogenesis in iron-deficient rats. Values are presented as median (interquartile range, IQR). *n* = 4 per group.

**Figure 6 nutrients-14-02063-f006:**
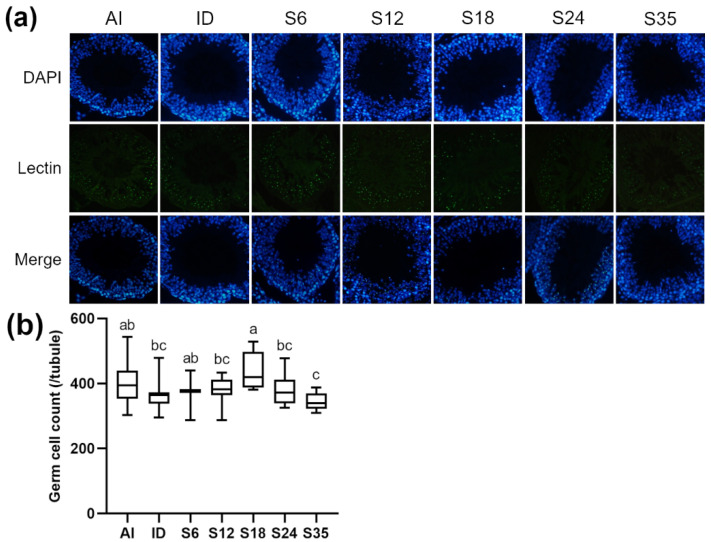
Effects of different doses of iron supplementation on (**a**) testicular seminiferous tubules by DAPI (nucleus) and Lectin (acrosome) staining, and (**b**) germ cell numbers in stage IV of spermatogenesis in iron-deficient rats. Values are presented as median (interquartile range, IQR). *n* = 4 per group. Values not sharing the same letters are significantly different at *p* < 0.05.

**Figure 7 nutrients-14-02063-f007:**
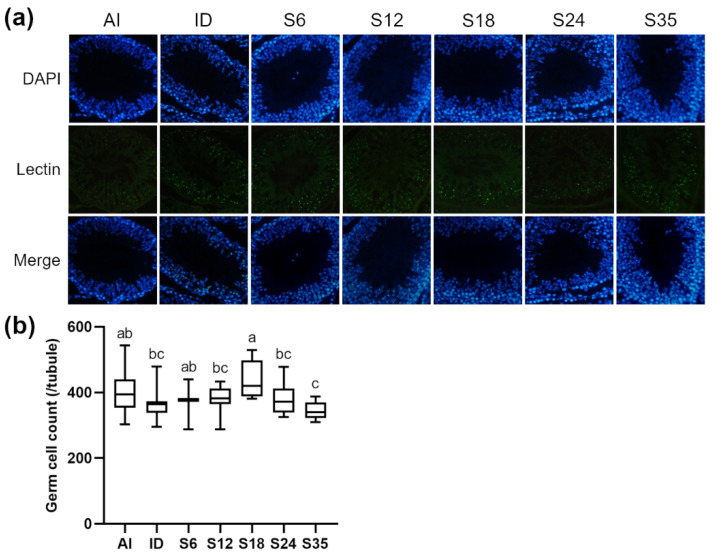
Effects of different doses of iron supplementation on (**a**) testicular seminiferous tubules by DAPI (nucleus) and Lectin (acrosome) staining, and (**b**) germ cell numbers in stage V of spermatogenesis in iron-deficient rats. Values are presented as median (interquartile range, IQR). *n* = 4 per group. Values not sharing the same letters are significantly different at *p* < 0.05.

**Figure 8 nutrients-14-02063-f008:**
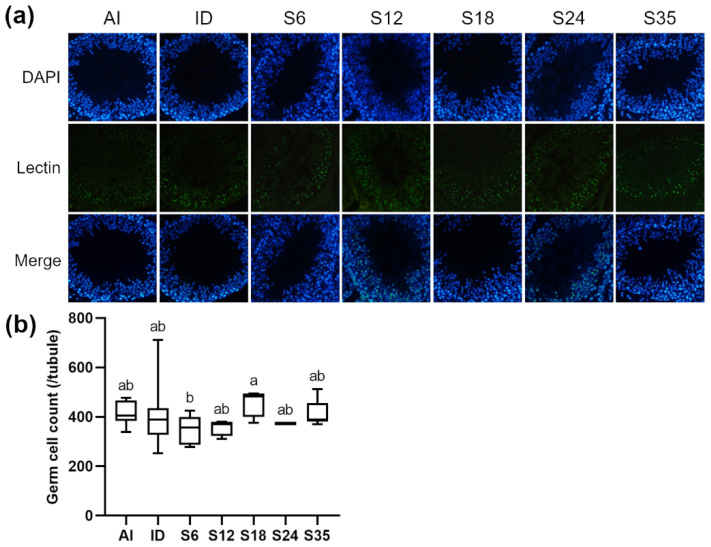
Effects of different doses of iron supplementation on (**a**) testicular seminiferous tubules by DAPI (nucleus) and Lectin (acrosome) staining, and (**b**) germ cell numbers in stage VI of spermatogenesis in iron-deficient rats. Values are presented as median (interquartile range, IQR). *n* = 4 per group. Values not sharing the same letters are significantly different at *p* < 0.05.

**Figure 9 nutrients-14-02063-f009:**
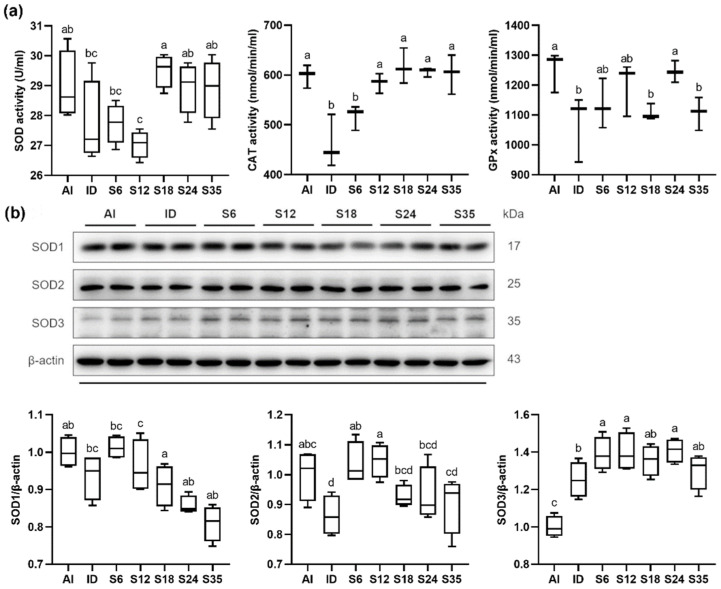
Effects of different doses of iron supplementation on (**a**) testicular antioxidants and (**b**) protein expression of SOD in iron-deficient rats. Values are presented as median (interquartile range, IQR). *n* = 4 per group. Values not sharing the same letters are significantly different at *p* < 0.05.

**Figure 10 nutrients-14-02063-f010:**
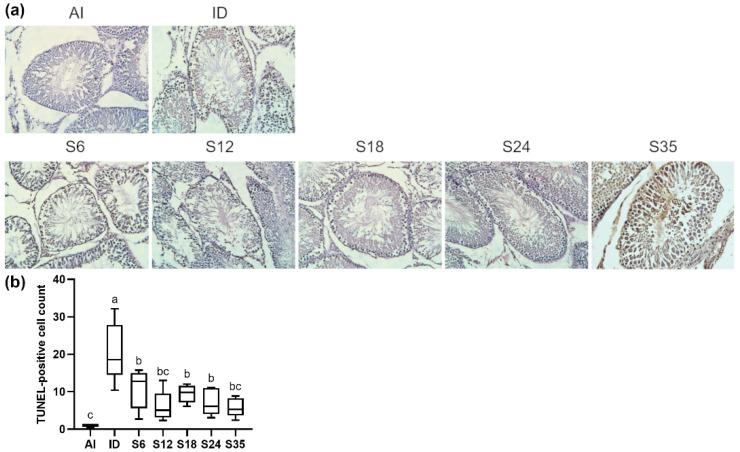
Effects of different doses of iron supplementation on (**a**) TUNEL staining of the testis and (**b**) TUNEL-positive cells in iron-deficient rats. Values are presented as median (interquartile range, IQR). *n* = 4 per group. Values not sharing the same letters are significantly different at *p* < 0.05.

**Figure 11 nutrients-14-02063-f011:**
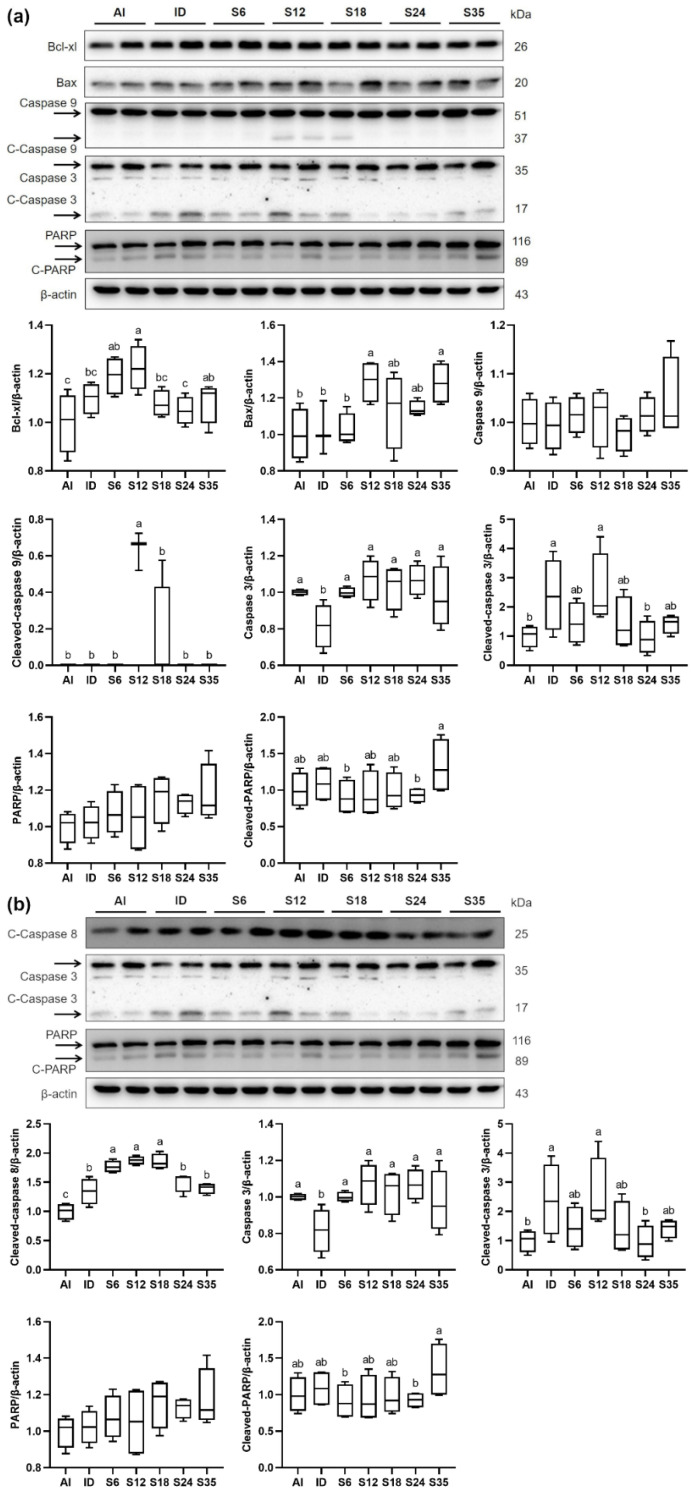
Effects of different doses of iron supplementation on protein expressions of markers related to the testicular (**a**) intrinsic apoptosis pathway and (**b**) extrinsic apoptosis pathway in iron-deficient rats. Values are presented as median (interquartile range, IQR). *n* = 3–4 per group. Values not sharing the same letters are significantly different at *p* < 0.05.

## Data Availability

Not applicable.

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
