# Peer review of "Effects of Iron Supplementation on Testicular Function and Spermatogenesis of Iron-Deficient Rats"

_nutrients, 2022, doi:10.3390/nu14102063_

Round 1
Reviewer 1 Report
The manuscript entitled „Effects of iron supplementation on testicular function and spermatogenesis of iron-deficient rats” presents interesting issue, but some issues should be corrected.
Abstract:
Brief justification of the study should be presented
Aim of the study should be presented
The numeric results of the study accompanied by the results of the statistical analysis should be presented.
Introduction:
While presenting the problem of iron deficiency anaemia, Authors should address gender-dependent differences and their reasons.
Materials and Methods:
The composition and nutritional value of applied diets should be presented
Authors should indicate if animals were housed together or separately and how it could have influenced experienced stress
It seems that Authors did not verify the normality of distribution of their data – they should do it and present the related methodology.
After verifying the normality of distribution, in case of parametric distribution mean ± SD should be presented, while for nonparametric distribution – median accompanied by minimum and maximum value.
The applied statistical test should be based on distribution
Results:
Authors should clearly present their results, as we even do not know if distribution is parametric or not
After verifying the normality of distribution, in case of parametric distribution mean ± SD should be presented, while for nonparametric distribution – median accompanied by minimum and maximum value.
The applied statistical test should be based on distribution
Instead of bar charts, Authors should present tables to be easier to follow by readers.
Discussion:
Authors should address within this section the form of iron, as well as factors potentially increasing and reducing its absorption.
Authors should not be focused on reproducing the results within this section.
Authors should: (1) compare gathered data with the results by other authors, (2) formulate implications of the results of their study and studies by other authors, (3) formulate the future areas which should be studied.
Authors should discuss all the potential limitations of the study (e.g. sample size)
Authors should briefly formulate conclusions.
Authors Contribution:
It seems that contribution of some Authors was only minor (YRL, TCC) and they did not participate in preparing manuscript. There is a serious risk of a guest authorship procedure which is forbidden. In such case they should be rather presented in Acknowledgements Section and not be indicated as authors of the study.
Author Response
Dear Professor Editor:
Please find enclosed our revised original paper entitled “Effects of iron supplementation on testicular function and spermatogenesis of iron-deficient rats”. We appreciated the comments and suggestions provided to further improve our manuscript.
Sincerely yours
Corresponding author: Chin-Yu Liu, Ph.D.
Nutritional Science, Fu Jen Catholic University, New Taipei City, Taiwan
No.510, Zhongzheng Rd., Xinzhuang Dist., New Taipei City 24205, Taiwan.
Telephone: +886-2-29053610
Fax: +886-2-29021215
e-mail: nf351.lab@gmail.com
Comments of Reviewer 1 to Author:
The manuscript entitled “Effects of iron supplementation on testicular function and spermatogenesis of iron-deficient rats” presents interesting issue, but some issues should be corrected.
Abstract:
Brief justification of the study should be presented
Aim of the study should be presented
The numeric results of the study accompanied by the results of the statistical analysis should be presented.
Our response:
We thank the reviewer for the helpful suggestion. We have added the brief justification “Iron deficiency is the most common micronutrient deficiency in the world. Previous studies have shown that iron deficiency increases oxidative stress and decreases antioxidant enzymes, and studies of male infertility indicated that oxidative stress may affect male reproductive functions.” and revised the description of statistical analysis methods “We presented as median (interquartile range, IQR) for continuous measurements and compared their differences using the Kruskal–Wallis test followed by the Mann–Whitney U test among groups. The results showed that as compared with the AI group, the ID group had a significantly lower serum testosterone and poorer spermatogenesis (The median [IQR] were 187.4 [185.6-190.8] of AI group vs.87.5 [85.7-90.4] of ID group in serum testosterone, p< 0.05; 9.3 [8.8-10.6] of AI group vs. 4.9 [3.4-5.4] of ID group in mean testicular biopsy score [MTBS], p< 0.05)” in the “Abstract” part.
Introduction:
While presenting the problem of iron deficiency anaemia, Authors should address gender-dependent differences and their reasons.
Our response:
We thank the reviewer for the professional opinion. According to the comment and combined the comments of another reviewer, we added the statement “According to the Nutrition and Health Survey in Taiwan 1993-1996 [18], 2.1% men was iron deficiency, and higher prevalence of iron deficiency was found in teenage boys and aged men. The reason that the iron-deficient group and five iron-supplemented groups with diets containing different doses of ferrous sulfate were designed, is to closely mimic the different status of iron deficiency.” in the “Discussion” part.
Materials and Methods:
The composition and nutritional value of applied diets should be presented
Authors should indicate if animals were housed together or separately and how it could have influenced experienced stress
It seems that Authors did not verify the normality of distribution of their data – they should do it and present the related methodology.
After verifying the normality of distribution, in case of parametric distribution mean ± SD should be presented, while for nonparametric distribution – median accompanied by minimum and maximum value.
The applied statistical test should be based on distribution
Our response:
We thank the reviewer for the academic suggestion. We have added the composition and nutritional values of feeding diet in the supplementary table. Meanwhile the rats of the study were housed in together since childhood (3- week-old), which have influenced less experimental competition stress. We eventually revised the presented results as median (interquartile range, IQR) for continuous measurements because of non‑normal distribution. In addition, we compared their differences using the Kruskal–Wallis test followed by the Mann–Whitney U test among groups. Please see the revised manuscript in “2.8. Statistical analyses” section of Materials and Methods.
Results:
Authors should clearly present their results, as we even do not know if distribution is parametric or not
After verifying the normality of distribution, in case of parametric distribution mean ± SD should be presented, while for nonparametric distribution – median accompanied by minimum and maximum value.
The applied statistical test should be based on distribution
Instead of bar charts, Authors should present tables to be easier to follow by readers.
Our response:
We thank the reviewer for the academic suggestion. We eventually revised the presented results as median (interquartile range, IQR) for continuous measurements because of non‑normal distribution. In addition, we compared their differences using the Kruskal–Wallis test followed by the Mann–Whitney U test among groups. Meanwhile the initial presented bar charts of figures were replaced by the box plot totally in the “Figures” part. Moreover we added the varied values of individual group of the study in the “Supplementary Tables” part for the readers easier to follow.
Discussion:
Authors should address within this section the form of iron, as well as factors potentially increasing and reducing its absorption.
Authors should not be focused on reproducing the results within this section.
Authors should: (1) compare gathered data with the results by other authors, (2) formulate implications of the results of their study and studies by other authors, (3) formulate the future areas which should be studied.
Authors should discuss all the potential limitations of the study (e.g. sample size)
Authors should briefly formulate conclusions.
Our response:
We thank the reviewer for the helpful comment. We have rewrote and revised the description of the Discussion according to the suggestions of reviewer.
Authors Contribution:
It seems that contribution of some Authors was only minor (YRL, TCC) and they did not participate in preparing manuscript. There is a serious risk of a guest authorship procedure which is forbidden. In such case they should be rather presented in Acknowledgements Section and not be indicated as authors of the study.
Our response:
We thank the reviewer for the opinion. YRL and TCC were the graduate students for the study processing and they participated in the preparing the revised manuscript. Herein we judged they should be listed in the coauthors not only presented in the Acknowledgements Section.

Reviewer 2 Report
In this rat model of diet-induced iron deficiency, reproductive functions including testosterone level, semen quality, testicular histology and spermatogenesis, in addition to possible modulation pathways, were explored.
The results demonstrated potential effects of iron deficiency on testis function and prospective improvements resulting from iron supplementation after iron deficiency.
The work has many limitations, which were recognized by the authors, but the work is good and the discussion is well structured to match the results.
There are still some points to be explained.
- Why did the authors use three-week-old rats, when sexual maturity is such an important aspect in studies of reproductive function? The age of rats referred to as 'adult' varied between six and 20 weeks [1].
- As shown in (Figure 4) demonstrating the mean seminiferous tubule diameter (MSTD) and mean testicular biopsy score (MTBS), the results showed unlike the normal structure observed in the AI group, testicular sections from rats exposed to a low-iron diet exhibited a thinner seminiferous epithelium, cavities, and scattered and decreased germ cells in seminiferous tubules; iron supplementation improved this damage. On the other hand in (Figures 5-8) the germ cell counts are for example in the ID group the same or even higher than the IA group. Please explain!
- What about the seminiferous tubule diameter (MSTD)! Did the results fit with those of Figure 4
- The green fluorescence in lectin staining images is not visible in Figure 5a-8a. Because of the images Quality, the description of the Testicular Sections page 7 lines 224-240 cannot be followed in these Figures .
- Define the characteristics of the stages II-VI using images with high magnification and good visible staining.
- In 2.1 define the Iron-supplementation doses S… PPM
- Define mean testicular biopsy score (MTBS)
- Labeling of the graphics must be enlarged
References
- Jackson, S.J.; Andrews, N.; Ball, D.; Bellantuono, I.; Gray, J.; Hachoumi, L.; Holmes, A.; Latcham, J.; Petrie, A.; Potter, P., et al. Does age matter? The impact of rodent age on study outcomes. Lab Anim 2017, 51, 160-169, doi:10.1177/0023677216653984.
Author Response
Dear Professor Editor:
Please find enclosed our revised original paper entitled “Effects of iron supplementation on testicular function and spermatogenesis of iron-deficient rats”. We appreciated the comments and suggestions provided to further improve our manuscript.
Sincerely yours
Corresponding author: Chin-Yu Liu, Ph.D.
Nutritional Science, Fu Jen Catholic University, New Taipei City, Taiwan
No.510, Zhongzheng Rd., Xinzhuang Dist., New Taipei City 24205, Taiwan.
Telephone: +886-2-29053610
Fax: +886-2-29021215
e-mail: nf351.lab@gmail.com
Comments of Reviewer 2 to Author:
In this rat model of diet-induced iron deficiency, reproductive functions including testosterone level, semen quality, testicular histology and spermatogenesis, in addition to possible modulation pathways, were explored.
The results demonstrated potential effects of iron deficiency on testis function and prospective improvements resulting from iron supplementation after iron deficiency.
The work has many limitations, which were recognized by the authors, but the work is good and the discussion is well structured to match the results.
There are still some points to be explained.
- Why did the authors use three-week-old rats, when sexual maturity is such an important aspect in studies of reproductive function? The age of rats referred to as 'adult' varied between six and 20 weeks [1].
Our response:
We thank the reviewer for the academic suggestion. According to the Nutrition and Health Survey in Taiwan 1993-1996, 2.1% men was iron deficient, and higher prevalence of iron deficiency was found in teenage boys and aged men. The three-week-old rats mimic the iron-deficient teenage boys, explored effects of iron-deficient happens since this period and whether iron depletion was recovered in different levels (5 supplemented groups) in adulthood would help restoring the negative impacts.
Reference
- Shaw, N.-S.; Yeh, W.-T.; Pan, W.-H. Prevalence of Iron Deficiency in the General Population in Taiwan. Nutritional Sciences Journal 1999, 24, 119-138, doi:10.6691/jcns.199902_24(1).0007.
- As shown in (Figure 4) demonstrating the mean seminiferous tubule diameter (MSTD) and mean testicular biopsy score (MTBS), the results showed unlike the normal structure observed in the AI group, testicular sections from rats exposed to a low-iron diet exhibited a thinner seminiferous epithelium, cavities, and scattered and decreased germ cells in seminiferous tubules; iron supplementation improved this damage. On the other hand in (Figures 5-8) the germ cell counts are for example in the ID group the same or even higher than the IA group. Please explain!
Our response:
We thank the reviewer for the professional comment. According to the mean seminiferous tubule diameter (MSTD) and mean testicular biopsy score (MTBS) of Figure 4, it showed the testicular sections from rats exposure to a low-iron diet exhibited a thinner seminiferous epithelium, cavities, and scattered and decreased germ cells in seminiferous tubules. The presented results corresponds to the revised Figures 5-8 box plots, the germ cell counts in the ID group are less than the AI group although without statistical significance.
- What about the seminiferous tubule diameter (MSTD)! Did the results fit with those of Figure 4
The green fluorescence in lectin staining images is not visible in Figure 5a-8a. Because of the images Quality, the description of the Testicular Sections page 7 lines 224-240 cannot be followed in these Figures.
- Define the characteristics of the stages II-VI using images with high magnification and good visible staining.
- In 2.1 define the Iron-supplementation doses S… PPM
Define mean testicular biopsy score (MTBS)
Labeling of the graphics must be enlarged
Our response:
We thank the reviewer for the academic comment. In Figure 4, the mean seminiferous tubule diameter (MSTD) of the ID group is more than the AI group but without statistically significant, however it displayed a thinner seminiferous epithelium, cavities, and scattered and decreased germ cells in seminiferous tubules. We have redone the staining images of Figures 5-8 to get the better quality. Moreover the description about the doses of Iron supplementation was added in 2.1 of “Materials and Methods”. Eventually the mean testicular biopsy score (MTBS) is an evaluation of the germ epithelial cell maturation following Johnsen’s criteria. The mean score of iron-deficient rats was 5.3, represented as no spermatozoa or spermatids but many spermatocytes observed in the seminiferous tubules.
References
- Johnsen, S.G. Testicular Biopsy Score Count – A Method for Registration of Spermatogenesis in Human Testes_Normal Values and Results in 335 Hypogonadal Males. Hormones 1970, 1, 2-25.
- Kobyliak, N.; Falalyeyeva, T.; Kuryk, O.; Beregova, T.; Bodnar, P.; Zholobak, N.; Shcherbakov, A.; Bubnov, R.; Spivak, M. Antioxidative effects of cerium dioxide nanoparticles ameliorate age-related male infertility: Optimistic results in rats and the review of clinical clues for integrative concept of men health and fertility. EPMA Journal 2015, 6, 12, doi:10.1186/s13167-015-0034-2.

Round 2
Reviewer 1 Report
The manuscript entitled „Effects of iron supplementation on testicular function and spermatogenesis of iron-deficient rats” presents interesting issue, but some issues should be corrected.
Abstract:
Aim of the study should be presented (e.g. “The aim of the study was…”) – not what was done, but what was intended to.
Introduction:
While presenting the problem of iron deficiency anaemia, Authors should address gender-dependent differences and their reasons. This issue should be presented in Introduction Section to present adequately background of the study.
Materials and Methods:
If animals were housed together Authors should indicate in the Materials and Methods Section how it could have influenced experienced stress
Discussion:
Authors should briefly formulate conclusions.
Author Response
Dear Professor Editor:
Please find enclosed our revised original paper entitled “Effects of iron supplementation on testicular function and spermatogenesis of iron-deficient rats”. We appreciated the comments and suggestions provided to further improve our manuscript.
Sincerely yours
Corresponding author: Chin-Yu Liu, Ph.D.
Nutritional Science, Fu Jen Catholic University, New Taipei City, Taiwan
No.510, Zhongzheng Rd., Xinzhuang Dist., New Taipei City 24205, Taiwan.
Telephone: +886-2-29053610
Fax: +886-2-29021215
e-mail: nf351.lab@gmail.com
Comments and Suggestions for Authors
The manuscript entitled „Effects of iron supplementation on testicular function and spermatogenesis of iron-deficient rats” presents interesting issue, but some issues should be corrected.
Abstract:
Aim of the study should be presented (e.g. “The aim of the study was…”) – not what was done, but what was intended to.
Our response:
We thank the reviewer for the helpful suggestion. We have revised the statement as “Aim of this study was to investigate the effects of iron supplementation on spermatogenesis and testicular functions in iron-deficient rats.” in the “Abstract” part.
Introduction:
While presenting the problem of iron deficiency anaemia, Authors should address gender-dependent differences and their reasons. This issue should be presented in Introduction Section to present adequately background of the study.
Our response:
We thank the reviewer for the professional opinion. We have formulated the statement of conclusions as “Human fertility is affected not only biological consequences but also so socioeconomic factors [16]. Declining fertility is a key driver behind the rapid aging of populations worldwide. Taiwan is also going on the demographic transition from high birth and death rates to low birth and death rates. Gender-dependent differences is one of the issues but women always take on disproportionate share of fertility tasks. Unlike female infertility, male infertility is not well reported in general. Thus, we want to explore the association of male reproductive health-related diseases and the potential role of nutrition factors. ” in the “Introduction” part.
Materials and Methods:
If animals were housed together Authors should indicate in the Materials and Methods Section how it could have influenced experienced stress
Our response:
We thank the reviewer for the academic suggestion. Following the way that animals in research should be spared unnecessary harm, pain, or distress, we performed the experiment in the qualified laboratory animal center and cared by the veterinary. Due to the daily intake have to be recorded, the rats were fed in the cage individually. We should describe clearly and revised the Materials and Methods “housed in together but kept in the cage individually under controlled conditions”
Discussion:
Authors should briefly formulate conclusions.
Our response:
We thank the reviewer for the academic suggestion. We have formulated the statement of conclusions as “In conclusions, iron deficiency modulated testicular anti-oxidative enzymes and protein expressions of apoptotic markers to affect testosterone biosynthesis-related enzymes and the serum testosterone level, and thus altered spermatogenesis and the morphology of the testis. Under iron supplementation, testicular functions were improved via increasing activities of testicular antioxidants.” in the “Conclusions” part.
